# Optimal Transport-Based Domain Alignment as a Preprocessing Step for Federated Learning

## Abstract

Federated learning is a subfield of machine learning that avoids sharing local data with a central server, which can enhance privacy and scalability. The inability to consolidate data in a central server leads to a unique problem called dataset imbalance, which is where agents in a network do not have equal representation of the labels one is trying to learn to predict. In FL, fusing locally-trained models with unbalanced datasets may deteriorate the performance of global model aggregation; this further reduces the quality of updated local models and the accuracy of the distributed agents' decisions. In this work, we introduce an Optimal Transport-based preprocessing algorithm that aligns the datasets by minimizing the distributional discrepancy of data along the edge devices without breaking privacy concerns. We accomplish this by leveraging Wasserstein barycenters when computing channel-wise averages. These barycenters are collected in a trusted central server where they collectively generate a target RGB space. By projecting our dataset towards this target space, we minimize the distributional discrepancy on a global level, which facilitates the learning process due to a minimization of variance across the samples in the analyzed network. We demonstrate the capabilities of the proposed approach over the CIFAR-10 dataset, where we show its capability of reaching higher degrees of generalization in fewer communication rounds.

## 1 Introduction

Federated learning (FL) is a subfield of machine learning (ML) that tackles decentralized, or distributed, learning. It lends itself as a solution to machine learning problems where datasets cannot be shared due to privacy concerns (McMahan et al., 2017) and, in turn, enhances scalability. Its privacy-centered approach led to its adoption in various fields, such as healthcare (Xu et al., 2021; Rieke et al., 2020). Another interesting paradigm tackled by FL is the personalization of models where the expected behavior varies from user to user in the network. In Tan et al. (2022), the authors perform a comprehensive dive over the challenges and use cases of personalized federated learning.

While FL has a myriad of applications, especially with the increasing regulations on personal data, its constraints must be carefully handled to ensure the stable training of models. These constraints include transmission rates, computational power at the edge, privacy, and dataset imbalance. The problem of imbalanced datasets, as can be seen in Li et al. (2020), arises when edge devices do not all have the same number of samples of each predicted class and/or have different training dataset sizes; this can equivalently mean they have different generating distributions. Similarly, the quality of the local data provides another obstacle in this setting (Wu et al., 2023). Both of these issues create imbalanced representations that worsen the learning process. Our goal is to tackle this imbalance problem by looking at it as an alignment of the different local generating distributions. To this end, we design a preprocessing mechanism that is model- and learning algorithm-agnostic and allows for a transformation of the local data to a distribution space comprised of information from all agents in the network.

## 2 RELATED WORKS

The issue of dataset imbalance, also known as the domain-distribution discrepancy problem or the multiple-source domain problem, has been a core challenge in federated learning. In this section, we will review various approaches to address this challenge. To our knowledge, our approach is the first of its kind in terms of performing a zero-shot alignment and the second to introduce optimal transport (OT) to perform a distribution alignment in FL. While it may be interesting to think of various preprocessing techniques to compare against, to our knowledge, none exist. In our research, we have only seen local alignments and processing techniques such as feature normalization (e.g., min-max scaling) or standardization (e.g., Z-score scaling). While these techniques are useful to help train models by scaling feature values to desirable ranges, they do not align samples across agents. The lacking literature further supports the importance of our idea.

The most relatable work we have found is that of Farnia et al. Farnia et al. (2022) where the authors introduce an OT-based alignment step during the process of training the model. Their work is comprised of an iterative approach to both computing the projection map and the space they are projecting to. The authors begin by extending the standard OT task between two distributions to a multi-marginal OT problem. They then use these results to create a min-max optimization problem which leads to their algorithm called *FedOT*. While *FedOT* introduces a dynamic approach to learning the necessary maps to transform the data, we instead bypass this step. Rather than iteratively learning a transport map and the target space, we instead compute the target space using a two step approach (described later as an RGB-wise Wasserstein barycenter) and then leverage Ferradans et al. (2014) which demonstrates how to compute the alignment map. Our approach introduces two benefits. First, by computing the target space in one shot rather than dynamically, we align images to a unified representation. Secondly, by simplifying the alignment process, we lower computation cost by removing the iterative process that is needed to learn the map which is then used to perform the alignment.

In Wang et al. (2022), Wang et al. comprehensively surveyed the field of domain adaptation. They describe three primary techniques used to solve this problem: data manipulation, representation learning, and learning strategy. Out of the diverse list of papers surveyed, few are directly solving the problem. First, Chen et al. (2020) focuses on personalization through transfer learning. The pseudo domain generalization occurs in the transfer step, as the *FedHealth* algorithm aligns the lower levels of the convolutional neural network aggregated in the cloud with those locally stored, and personalized, in each agent's device. On the other hand, in Wu & Gong (2021), Wu and Gong tackle the learning strategy approach of domain generalization (DG) and unsupervised domain adaptation (UDA). In their algorithm, called Collaborative OPtimization and Aggregation (COPA), each agent in the network contains a local dataset, a feature extractor, and a classifier. The local feature extractors are collected to create a global feature extractor that is domain-invariant. The local classifiers are ensembled to create the global classifier. These global models are used to update the local models. After various iterative updates, COPA was able to converge to a point where it was comparable with state-of-the-art algorithms that focused on DG and UDA. Lastly, Zhang et al. (2021) subscribe to the "learning strategy" approach to solving domain generalization. They accomplish this through their algorithm *FedADG*, which uses an adversarial component to measure and then align different source distributions to a globally known, shared distribution. Similarly to COPA, their algorithm also focuses on learning a domain-invariant feature. While *FedADG* improves on some test cases, the authors mention their approach to generating invariant features yields results that are typically worse than an OT-based approach called L2A-OT (Zhou et al., 2020).

Thus far, we have covered papers that are directly related to our work. We now focus on other research that indirectly tackles, or at least provides, a direction to work with imbalanced datasets. We first turn to clustering in federated learning. In Sattler et al. (2020), Sattler et al. propose their main algorithm, clustered federated learning (CFL), which relies on a generalization of the FL problem: "There exists a partitioning of the clients such that all clients in each partition satisfy the conventional FL problem." The "conventional problem" implies there exists one model that can fit all data distributions across the network. In an imbalanced dataset paradigm, the agents' data-generating distributions are different. To overcome the inability of FL to handle this problem, CFL clusters the incoming gradients of the agents through a recursive cosine similarity-based bipartitioning and then updates the model's parameters once for each cluster by averaging the gradients of the clients in each cluster respectively.

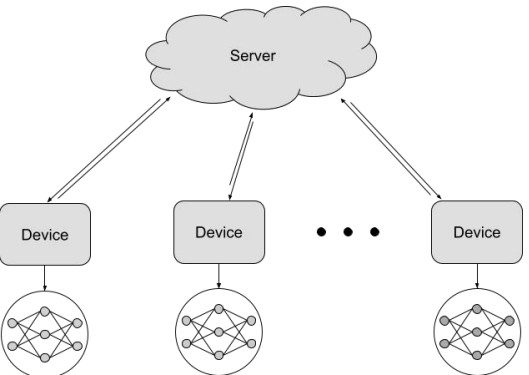

Figure 1: Federated learning architecture.

Moreover, in Hsu et al. (2020), Hsu et al. explore dataset imbalance in the unique datasets they created which allowed them to analyze the degree of imbalance based on geo-location tagging and distributional statistics. To tackle the class distribution shift problem in FL, they began with, *FedAvg*, short for federated averaging (McMahan et al., 2017), and updated it to include variance mitigation and a resampling strategy via adding weights to each agent. Their algorithms, respectively called *FedVC* and *FedIR*, computed the weights as a ratio of the target joint distribution of images and class labels and a sampled joint distribution of a specific client's image-label pair. The authors propose the assumption that the conditional distribution of an image sample $x$ and its label $y$ is equivalent, or a close approximate, to each agent's conditional; this assumption allowed through Bayes theorem to reformulate the weight function as a ratio of the global distribution of class $y$ and the agent's local distribution of class $y$. The weights are used to scale the updates of the global model sitting in the trusted server and lead to better accuracy.

Lastly, personalization is an interesting direction because it has been both used as a solution to the aforementioned problem and also as an application to improve local, customized, solutions for FL. Fallah et al. present *MAML* an algorithm that trains an "initial shared model" which is updated using a few gradient steps to yield a good, personalized model (Fallah et al., 2020). For more algorithms regarding personalization, we refer readers to Tan et al. (2022).

## 3 PRELIMINARIES

### 3.1 FEDERATED LEARNING

The goal of federated learning (FL) is to train a single (global) model that can make accurate predictions across all agents in a network. A typical FL architecture can be seen in Figure 1. FL problems contain constraints, some of which include: computational power at the edge, communication bandwidth, and data heterogeneity. Privacy is one of the key motivations for using FL which prevents the amalgamation of data in a central server and the use of traditional techniques. To overcome this problem, FL research introduced novel techniques that are typically a variation of parameter averaging or gradient averaging. A more comprehensive look at the challenges of FL and associated solutions can be found in Li et al. (2020). Independently of the approach, at its core, FL looks for a solution to

$$\min_w f(w), \quad f(w) = \sum_{a=1}^{A} p_a f_a(w) \tag{1}$$

where $A$ is the number of devices in our network, $p_a \geq 0$, and $\sum_a p_a = 1$. Each agent has a local objective, $f_a(w)$, which is averaged together as the global objective $f(w)$. For more details into equation 1, we refer readers to Li et al. (2020) and Imteaj et al. (2021).

## 3.2 Optimal Transport

Optimal Transport (OT), also known as the Monge-Kantorovich problem, is a mathematical framework that addresses the problem of finding the most cost-efficient way to transport mass from a source distribution to a target distribution (Torres et al., 2021). OT was originally formulated by Gaspard Monge in the 18th century (Monge, 1781). In the 20th century, the problem was refined by soviet mathematician Kantorovich and has since found widespread applications in various fields, including economics, computer vision, machine learning, and statistics. At its core, OT seeks to quantify the discrepancy between two probability distributions by defining a distance metric that considers both the magnitudes of the masses being transported and the distances over which they are moved. Unlike traditional distance metrics, such as Euclidean distance, which focus solely on point-to-point comparisons, OT provides a more geometrically nuanced measure that captures the structural similarities between distributions.

We begin by looking at Monge's original problem over discrete spaces for simplicity, but continuous equivalents exist. Given discrete measures $\alpha, \beta$, the *Monge Map* is given by

$$\min_T \left\{ \sum_{i=1}^{n} c(x_i, T(x_i)) \quad : \quad T_\# \alpha = \beta \right\} \tag{2}$$

Monge's problem is asking for a surjection from the source distribution to the target distribution, as can be seen by the restriction on $T$, a push-forward operator. Simply said, $T$ ensures mass is preserved during the transportation. While this formulation is simple to read, it is extremely difficult to solve due to non-convexity and degeneracy. Kantorovich sought to fix this by introducing the concept of mass-splitting. Where Monge required all the mass from any point in the domain to be mapped to one point in the target distribution, mass-splitting allows the mass from the domain to be broken up and mapped to different locations in the codomain. The OT problem, now called the Monge-Kantovorich problem, reads as follows:

$$L_{\mathbf{C}}(\mathbf{a}, \mathbf{b}) = \min_{P \in U(a,b)} \langle \mathbf{C}, \mathbf{P} \rangle = \sum_{i,j} \mathbf{C}_{i,j} \mathbf{P}_{i,j}, \tag{3}$$

given the set of *admissible couplings*

$$U(\mathbf{a}, \mathbf{b}) = \left\{ P \in R_+^{n \times m} : \mathbf{P} \mathbb{1}_m = \mathbf{a}, \mathbf{P}^T \mathbb{1}_n = \mathbf{b} \right\}$$

Intuitively, one is looking for a permutation matrix $\mathbf{P}$ that determines how to distribute mass in a cost-minimizing fashion given the transportation cost $\mathbf{C}$.

OT presents a novel approach to comparing two probability distributions. With additional constraints, OT yields a metric, or a distance function, called the *Wasserstein metric*, or *Earth Mover's Distance*. Suppose that for some $p \geq 1$ and $\mathbf{C} = \mathbf{D}^p$ where $\mathbf{D} \in \mathbb{R}^{n \times n}$ is a distance on $[\![n]\!]$,

1. $\mathbf{D}$ is symmetric,
2. $\text{diag}(\mathbf{D}) = \mathbf{0}$,
3. $\forall (i, j, k) \in [\![n]\!]^3, \mathbf{D}_{i,k} \leq \mathbf{D}_{i,j} + \mathbf{D}_{j,k}$

then the p-Wasserstein distance is

$$W_p(\mathbf{a}, \mathbf{b}) = L_{\mathbf{D}^p}(\mathbf{a}, \mathbf{b})^{1/p}. \tag{4}$$

Equipped with a distance function, we can now define an averaging function, called a Wasserstein barycenter (WB):

$$\min_{\mathbf{a} \in \Sigma_n} \sum_{s=1}^{S} \lambda_s W_p^p(\mathbf{a}, \mathbf{b}_s), \tag{5}$$

where $\lambda_s$ is a real-valued weight (usually a uniform distribution such that each input probability vector is given an equal amount of importance Cuturi & Doucet (2014)) and $\Sigma_n$ is a probability

simplex with $n$ bins. Intuitively, given a set of input probability vectors $\mathbf{b}_s$, we are looking for a probability vector $\mathbf{a}$ that minimizes the weighted sum of the p-Wasserstein distance between $\mathbf{a}$ and each $\mathbf{b}$.

There are different methods to solve the optimal transport problem and to compute Wasserstein barycenters. The current state-of-the-art methods rely on entropic regularization. In Cuturi (2013), Marco Cuturi introduces a solution to quickly solve the entropy regularized OT problem, $W_{reg}(\mathbf{a}, \mathbf{b})$, using Sinkhorn's algorithm, which now reads: For $\lambda > 0$,

$$d_M^\lambda(a, b) = \langle P^\lambda, C \rangle, \tag{6}$$

where

$$P^\lambda = \operatorname{argmin}_{P \in U(a,b)} \langle P, C \rangle - \frac{1}{\lambda} h(P).$$

Equipped with a computational method to solve the OT problem, Benamou et al. leveraged iterative Bregman projections to compute the entropic regularized Wasserstein barycenter, which can be seen in Benamou et al. (2015), leading to a faster and more general solution. The entropic regularized barycenter problem can be written as an extension of equation 5:

$$\min_{\mathbf{a} \in \Sigma_n} \sum_{s=1}^{S} \lambda_s W_{reg}(\mathbf{a}, \mathbf{b}_s) \tag{7}$$

where $\boldsymbol{\lambda} = \{\lambda_s\}_{s=1}^{S} \in \Sigma_S$.

## 4 OPTIMAL TRANSPORT-BASED PREPROCESSING

In this section, we introduce the preprocessing step that minimizes the distributional discrepancy in our network. We achieve this distribution-alignment goal by generating a target space to which we project all local data. The target space is generated from all data without losing privacy because WBs obfuscate the data in an irreversible fashion (more clarifications on the prevention of privacy leakage can be seen in A.1. Our proposed method has two main steps: the creation of the target space, and the projection step to perform alignment.

To create a relevant target space to align images to, it needs to contain information from all the agents in the network. We accomplish this in a two-step fashion. First, we compute representations of the local data through a channel-wise Wasserstein barycenter of the local images. The approach requires separating each local image its three color channels, grouping them by the respective color channels, and lastly computing the WB for each channel. Figure 2 demonstrates this workflow. In this work, we are using colored images, therefore, channel-wise implies red, green, and blue channels. The local computations produce an RGB-triplet called the local WBs. Next, the second step is to generate the target space by aggregating all local WBs in a central server, and repeating the same process of computing channel-wise barycenters; this yields the RGB-tripled called the global WB, or our target space. Readers will notice the steps just explained cover steps one and two of Figure 3.

The final steps of our preprocessing algorithm are steps three and four in Figure 3. First, we broadcast the global barycenters to the agents in the network. Then, we align the local images to target space by projecting them to the global WB. The projection process is composed of computing transportation plans to the target space that allows the color channels of the original images to be transferred and aligned, similarly to computing color transfer maps or domain adaptation in Ferradans et al. (2014); Courty et al. (2016) respectively. The entire preprocessing steps just described is explicitly laid out in algorithm 1. For more in-depth information on the implementation details of the steps just described, we refer readers to A.2.

Lastly, there are various custom variable names in the algorithm 1. To facilitate going through the algorithm, we will define them here. $\mathcal{WB}_r^a$, $\mathcal{WB}_g^a$, and $\mathcal{WB}_b^a$ are the Wasserstein barycenters of agent $a$ for channels red, green, and blue respectively. The terms $\mathbf{Img}_i^{red}$, $\mathbf{Img}_i^{green}$, and $\mathbf{Img}_i^{blue}$ imply, respectively, the red, green, and blue channels of the $i^{th}$ image. Furthermore, $\mathcal{WB}^G$ is the target space (the global Wasserstein barycenter) composed of the global RGB-triplet.

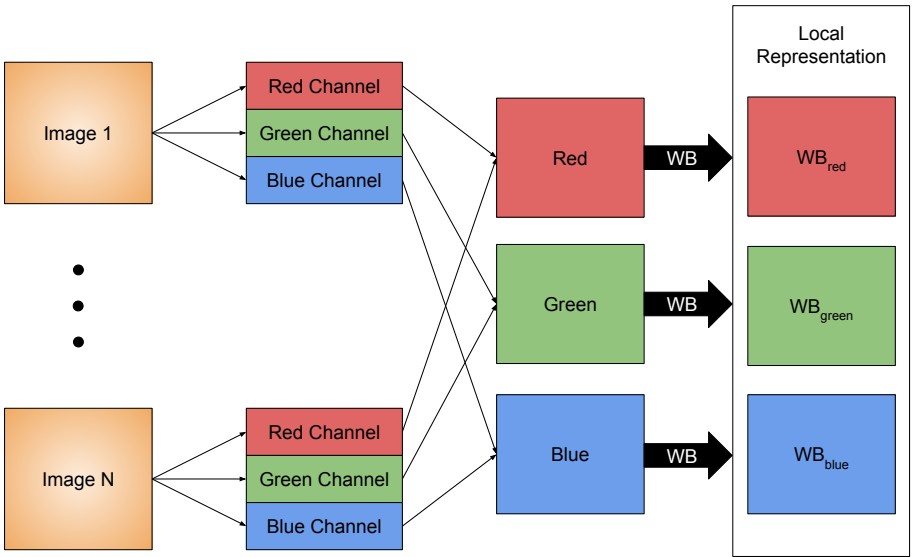

Figure 2: Channel-wise WB of local images.

---

**Algorithm 1** OT-based preprocessing

---

**Definitions:** Let $\mathbf{B} = \{\mathbf{b}_s\}_{s=1}^{S}$ such that $\mathrm{WB}(\mathbf{B})$ is the solution to equation (7).

---

    **For each agent** $a = 1, 2, \ldots, N$
        $\mathbf{R} = \{\mathbf{Img}_i^{red}\}_{i=1}^{M}$
        $\mathbf{G} = \{\mathbf{Img}_i^{green}\}_{i=1}^{M}$
        $\mathbf{B} = \{\mathbf{Img}_i^{blue}\}_{i=1}^{M}$
        $\mathcal{WB}_r^a = \mathrm{WB}(\mathbf{R})$
        $\mathcal{WB}_g^a = \mathrm{WB}(\mathbf{G})$
        $\mathcal{WB}_b^a = \mathrm{WB}(\mathbf{B})$
        Distribute $\mathcal{WB}_r^a, \mathcal{W}_g^a, \mathcal{WB}_b^a$ to a central server
    $\mathbf{R}_G = \{\mathcal{WB}_r^a\}_{a=1}^{N}$
    $\mathbf{G}_G = \{\mathcal{WB}_g^a\}_{a=1}^{N}$
    $\mathbf{B}_G = \{\mathcal{WB}_b^a\}_{a=1}^{N}$
    $\mathcal{WB}_r^G = \mathrm{WB}(\mathbf{R}_G)$
    $\mathcal{WB}_g^G = \mathrm{WB}(\mathbf{G}_G)$
    $\mathcal{WB}_b^G = \mathrm{WB}(\mathbf{B}_G)$
    $\mathcal{WB}^G = \{\mathcal{W}_r^G, \mathcal{WB}_g^G, \mathcal{WB}_b^G\}$
    Distribute $\mathcal{WB}^G$ to **all** agents
    **For each agent** $a = 1, 2, \ldots, N$
        **For each image** $i = 1, 2, \ldots, M$ **of agent** $a$
            Project image $i \rightarrow \mathcal{WB}^G$
**Output:** Local datasets are transformed through the projections and are ready to be used for learning given any FL algorithm

---

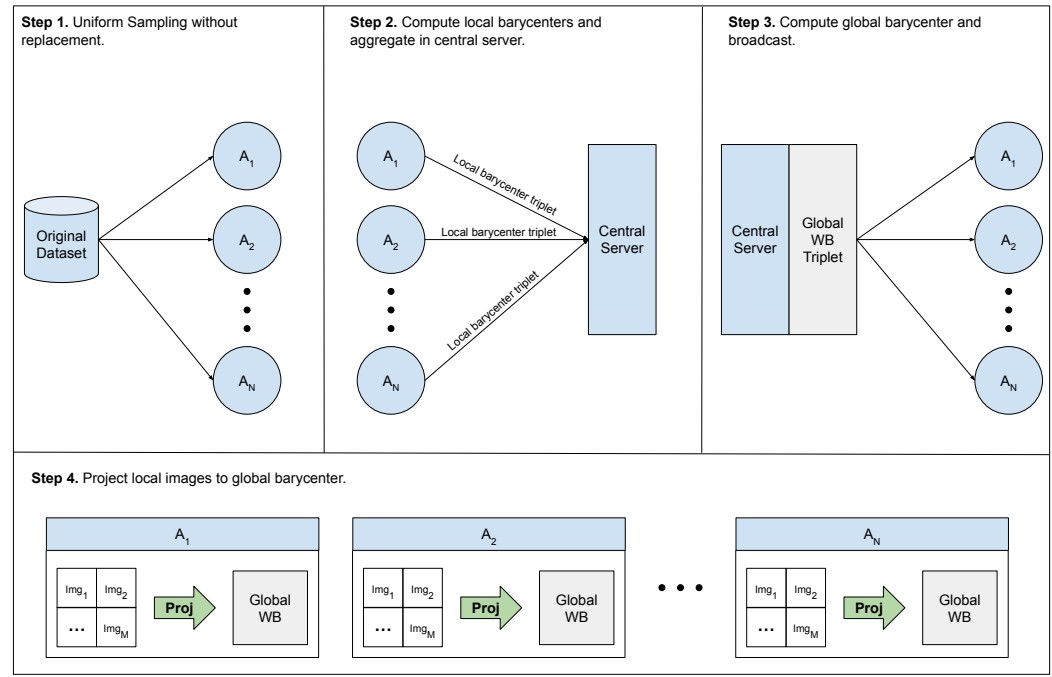

Figure 3: This image is a visual representation of the preprocessing algorithm given a network with $N$ agents which contain $M$ local images.

## 5 EXPERIMENTS AND RESULTS

The goal of our paper is to demonstrate the functionality and utility of our preprocessing step in federated learning. Our framework is built independently of the learning algorithm, which allows for flexible integration into any FL pipeline that may have different goals (e.g., increase privacy through differential privacy solutions to FL). Nonetheless, to demonstrate the advantages of using our preprocessing step, we require a learning algorithm. To this end, we chose to work with federated averaging, *FedAvg*, which trains local algorithms and performs parameter averaging to aggregate the local models into a global model. To show the improvements our method provides to the learning process, we designed a few experiments. Each experiment begins with the following common setup steps. First, we design the network by choosing the number of agents that make up our network. Next, we initialize their models identically (we refer readers to A.2 for implementation details such as hyperparameters). Starting with equivalent initial conditions is currently a requirement for FL, otherwise averaging these models can yield "arbitrarily bad models" (McMahan et al., 2017; Goodfellow et al., 2015). The base model architecture is a convolutional neural network (CNN) with roughly one million parameters and can be seen in Figure 5. Next, we distribute the data randomly by uniformly sampling, without replacement, images. By sampling uniformly without replacement, we ensure that each agent has varying amounts of data and are not completely homogenous (meaning each agent has equal number of images per class), which introduces additional difficulties during training. With the data distributed to the edge devices, we use our preprocessing algorithm (1) to align them. Once the data is aligned, we use *FedAvg* to train a global model.

There are various ways of deciding when to perform the synchronization step, which is where parameter averaging happens. Similar to other works, we run more than one epoch per synchronization step. On one hand, choosing one epoch per accumulation step did not induce sufficient local training. Conversely, too many local epochs resulted in local overfitting and yielded worse global outcomes. To demonstrate complete results, we ran simulations where all agents contributed to learning the global model and also ran simulations where a random selection of agents contributed (the agents were sampled according to a uniform distribution). As per convention, this is labeled $N/P$ where $N$ is the total number of agents in the network and $P$ is the number of agents that participate in training the global model. In experiments where $N = P$, we used two local training epochs per

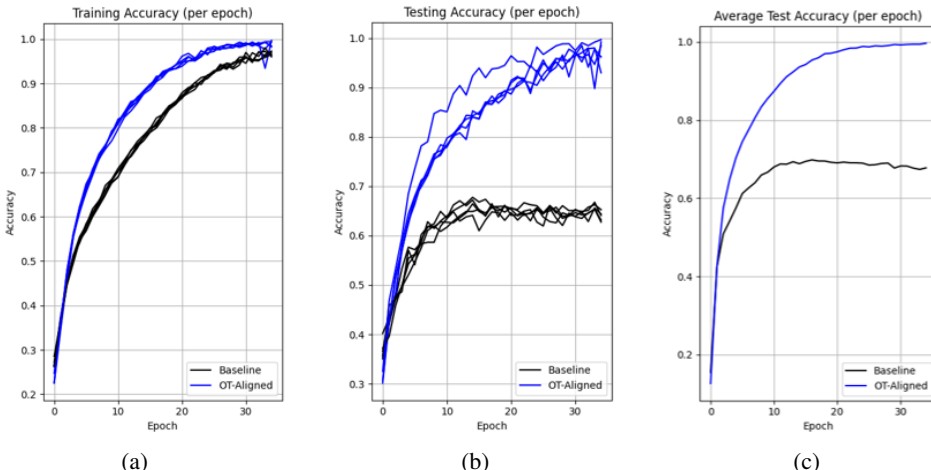

(a)                    (b)                    (c)

Figure 4: The images represent the results of a simulation with 10 agents and using all 10 agents to train the global model. The graphs are made by sampling 5 of the 10 agents to show their respective local models and the final global model. In all graphs, we plot a baseline of *FedAvg* without alignment and *FedAvg* with our "OT-aligned" preprocessing. (a) Accuracy of local models on training data. (b) Accuracy of local models on test data. (c) Accuracy of the global model on test data.

synchronization step as it resulted in the smoothest global accuracy curve. When $P << N$, the number of samples in the local dataset becomes extremely small, closer and closer to the batch size. Therefore, we need to increase the number of local epochs to ensure sufficient training. For these scenarios, we raised the local epoch count to five. The results of our simulations are shown in tables 1.

In table 1, the first column shows the number of agents in the network and the second shows the number of synchronization steps required to achieve the respective accuracy score. One thing to note is the increasing number of epochs as we increased $N$, the number of clients. To ensure our results showed the maximum potential of both algorithms, ours and traditional *FedAvg*, we aimed to train until convergence, which was only possible by training the models for longer periods. While the number of communication rounds seems quite large, there are two facts readers should take into consideration. First, our model is smaller than used in other work. While this choice was arbitrary and meant to be a placeholder to ensure proper simulations, early experimental results showed high accuracy scores and presented an, originally unknown, additional benefit. Second, we have demonstrated the best generalization of current standing work. Moreover, our choice of learning algorithm, *FedAvg* is quite simple, and therefore pairing our preprocessing algorithm with more efficient learning techniques will only serve to improve convergence speed while maintaining higher accuracy. As we can see in table 1, *FedAvg* with our preprocessing step yields higher accuracy across the board. For a closer inspection, we graphed the results for the simulation with $N = P = 5$ in Figure 4. Figure 4a shows a plot of the local models' training accuracy for both *FedAvg* without OT-preprocessing, labeled the "Baseline," and OT-preprocessed, labeled "OT-Aligned." Figure 4b plots the testing accuracy of the local models. Lastly, Figure 4c demonstrates the results of the respective global model on the testing dataset under both paradigms. These plots also demonstrate a faster rate of convergence which can be seen given all choices of $N, P$ in our experiments. While we have thus far shown results for a custom CNN, we provide further explanations for this choice, along with additional simulations using a ResNet, in A.3.

Lastly, table 2 shows a comparison of our algorithm with other approaches. The results for a non-iid dataset paradigm for these various algorithms can be found, respectively, in the cited work column, or collectively in Luo et al. (2021) where the authors compile results and compare them against their proposed method (CCVR). Our OT-preprocessing algorithm, paired with *FedAvg*, an algorithm known for its simplicity, yielded the best results. Our simulations, while not using the exact same hyperparameters, are undoubtedly comparable since it is simpler, uses a significantly smaller model, and yields higher accuracy. These facts open the door for improvement via the merging of our

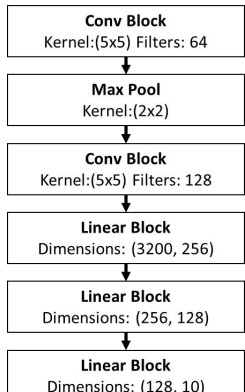

Figure 5: Convolutional neural network architecture used in our experiments.

Table 1: This table contains a comparison between *FedAvg* with and without our OT-preprocessing algorithm where all agents contributed to training the global model. The first column of the table demonstrates the number of clients in the simulated network (N) and the number of clients used to train the global model (P). The second column is the number of synchronization steps.

| Client (N/P) | Communication Rounds | *FedAvg* + **OT-Preprocessing** | *FedAvg* |
|---|---|---|---|
| 5/5 | 35 | **99.62** | 71.22 |
| 10/10 | 70 | **99.33** | 70.48 |
| 20/20 | 200 | **99.36** | 69.89 |
| 50/50 | 250 | **97.84** | 69.73 |
| 100/100 | 500 | **94.95** | 65.01 |
| 10/5 | 100 | **99.85** | 71.34 |
| 20/10 | 150 | **97.5** | 65.66 |
| 50/10 | 500 | **95.04** | 66.45 |
| 100/10 | 1000 | **93.34** | 66.16 |

work with algorithms that are tailored for other applications. For example, differential privacy-based methods that focus more on privacy may be able to increase both their privacy levels while increasing utility when paired with our algorithm. The methods listed in our comparison table are also applied in heterogeneous .

## 6 COMPLEXITY ANALYSIS

An important tradeoff to consider with our algorithm is the additional time that must be paid to convert the original dataset into an aligned dataset. As our method does not affect the learning algorithm, we only analyze the complexity of computing the barycenters and projecting the local data to the global barycenter, which is done prior to training. Therefore, this is an additional cost paid in addition to training. First, Kroshnin et al. (2019) gives us a complexity analysis for the iterative Bregman projection method of computing regularized barycenters, as introduced by Benamou et al. (2015). Given $n$ samples of dimension $d$ and regularization parameter $\epsilon$, we have a complexity of $O(nd^2/\epsilon^2)$. Moreover, the projection of the local images onto the global barycenters requires solving the OT problem. Cuturi (2013) demonstrates an empirical complexity of $O(d^2)$ with respect to the input dimension $d$. The complexity of our preprocessing is a combination of these two, contingent on the number of agents. Assume we have $N$ agents in our network, each with $M$ images. We must compute $N$ barycenters. Since these are computed in parallel, the time complexity is equivalent to computing one, with the addition of computing the global barycenter. Therefore, we have $O(Md^2/\epsilon^2)$ for local barycenters and $O(Nd^2/\epsilon^2)$ for the global barycenter. The projection time complexity is scaled in accordance with $M$, not $N$, as each agent projects in parallel. Therefore, the overall time complexity for our preprocessing algorithm is $O(Md^2/\epsilon^2)+O(Nd^2/\epsilon^2)+O(Md^2)$.

Table 2: This table compares the results of various approaches to federated learning. The first two rows are direct results from our experiments. The subsequent rows are results obtains from the respective papers. Readers should look at the cited work for implementation details which lead to the varying accuracy scores for the same algorithms (as can be seen below).

| Reference | Method | Accuracy |
|---|---|---|
| Our work | *FedAvg* + **OT-Preprocessing** ($N = 100, P = 10$) | **93.34** |
| (McMahan et al., 2017) | *FedAvg* ($N = 100, P = 10$) | 66.16 |
| (Li et al., 2021) | MOON | 69.1 |
| (Li et al., 2021) | FedAvg | 66.3 |
| (Li et al., 2021) | FedProx | 66.9 |
| (Li et al., 2021) | SCAFFOLD | 66.6 |
| (Li et al., 2021) | SOLO | 46.3 |
| (Wang et al., 2020) | FedMA | 87.53 |
| (Wang et al., 2020) | FedProx | 85.32 |
| (Wang et al., 2020) | FedAvg | 86.29 |
| (Luo et al., 2021) | FedAvg (CCVR) | 71.03 |
| (Luo et al., 2021) | FedProx (CCVR) | 70.99 |
| (Luo et al., 2021) | FedAvgM (CCVR) | 71.49 |
| (Luo et al., 2021) | MOON (CCVR) | 71.29 |

If the local number of images vary per agent, the worst-case complexity is the same and requires only setting $M = \mathrm{argmax}_i M_i$ for agents $i = 1, \ldots, N$.

## 7 CONCLUSION

In this work, we demonstrate the ability of our preprocessing algorithm to improve the convergence speed and generalization of traditional federated learning. We accomplish this by projecting local data to a space that encodes all local data. The projection to the same local space minimizes the distributional discrepancy between agents given a heterogeneous distribution of data. After aligning the local datasets, we train a model using *FedAvg*; this is an arbitrary choice, allowing any learning algorithm to be paired with our method. Our simulations show superior results than *FedAvg* and other comparable work, all of which do not contain preprocessed datasets. In comparison with the original work of McMahan et al. (2017), not only do we reach higher accuracy scores, but we do so with a model of fewer parameters. Our algorithm can be leveraged in any FL paradigm and paired with any learning algorithm as its functionality is in preprocessing the data to minimize the distribution discrepancy among different agents. Our work opens the door for future work where our preprocessing algorithm is fused with already existing learning-based algorithms. These potential fusions can lead to faster convergence and better generalization of already existing methods.

## 8 FUTURE WORK

In this work, we a presented novel idea for preprocessing colored images (i.e. CIFAR-10) as a way of aligning the various different generating distributions induced by federated learning. The improvements demonstrated by our RGB-wise Wasserstein barycenter technique leads to question of where else this framework can be applied. In future work we plan on exploring larger datasets (e.g., ImageNet), other data modalities, and other FL algorithms. While an RGB-wise approach is a technique for transforming images, we aim to develop new OT-based methods for transforming temporal data. Furthermore, we aim to determine the extent of the benefits our approach in other FL paradigms. For example, one question we aim to answer is if preprocessing data in the fashion described above may help retain the accuracy of models trained with higher noise in different privacy algorithms. While not an exhaustive list, these are some of the questions we aim to answer in our future research.

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

# A    APPENDIX

## A.1    PRIVACY CONCERNS

The concept of privacy is at the heart of federated learning. When introducing a novel approach that shares information between different devices, this could open an attack vector for the recreation of the original data. For example, when training a model using Federated Averaging over the gradients of local models, Geiping et al. (2020) show that the local images can be reconstructed. With this in mind, we asked ourselves about the vulnerability of computing the Wasserstein barycenters in the fashion described. To compute a Wasserstein barycenter, we must solve several optimal transport problems. Once solved, this mapping from probability distributions to the barycenter is not invertible; reasons for non-invertibility include information loss when solving OT problems with entropy-regularized formats and non-uniqueness of the solution of WBs. Furthermore, the RGB-wise barycenters do not share information across the color channels, making it further impossible to regenerate any source images. Therefore, distributing Wasserstein barycenters that are products of non-invertible optimization problems ensures the privacy of the local agents' data is not broken.

## A.2 Implementation Details

### A.2.1 Details on Training the CNN

While some of the details have been mentioned in the body of the text, we reiterate some details here and clarify others. First and foremost, for simulations where $N = P$ (e.g., 100/100), we ran two local training epochs before performing an accumulation step. When $N \neq P$, we increase the local training epochs to five. Regardless of which scenario we are simulating, we use a batch size of 16. Moreover, all simulations start by initializing a custom CNN, described in 5, and ensuring that all agents' models have the same parameter values. Similarly, all agents use the Adam optimizer with a learning rate of $1e^{-3}$ and the default beta values provided by PyTorch (betas=(0.9, 0.999)).

### A.2.2 Optimal Transport Details

It is important to clarify the details regarding the algorithms used to solve the OT problem and to compute Wasserstein barycenters. In this section of the appendix we will cover the details needed to implement the parts of our preprocessing framework that use OT. First and foremost, the core of our algorithm relies on computing the Wasserstein barycenter efficiently Benamou et al. (2015). We leverage the python optimal transport package Flamary et al. (2021) which contains a solver for the Bregman projection for regularized optimal transport. These methods require a cost function and a regularization term. We use a quadratic cost function and a regularization term equal to $1e - 2$. On another note, a technique that can be used to reduce the computational requirement is to sample a subset of pixel from each image and computer barycenters on this subset. In our work, we uniformly sampled 250 pixels from the input images, which greatly reduced computational time (the difference can be measured analytically using the time complexity shown in section 6).

Next, we will cover the details related to the projection step. To accomplish this, we leverage the work of Ferradans et al. (2014); Courty et al. (2016) and the python optimal transport package. With the global Wasserstein barycenter computed, we solve the entropic-regularized transport problem using the Sinkhorn algorithm; this can be accomplished using the *SinkhornTransport* method of python package. Similar to computing the barycenters, it is possible once again to select a subset of the pixels to compute the transportation map. For this step we used a regularization parameter of $1e - 1$ and once again selected a subset of 250 pixels. With the transport model in hand, we can transform the full image, yielding our transformed sample which will be used for training. The mathematical details of these algorithms are covered in detail in the aforementioned papers.

## A.3 Additional Simulations

The results shown in the body of the work were obtained using a custom CNN model, described in 5. This CNN has a simple architecture that was initially a placeholder during the development of our pipeline. During initial experimentation, results showed a surprising latent benefit of our preprocessing work. It is self-evident that if you simplify a problem, finding its solution is also simpler (e.g. using a kernel trick to linearize a non-linear problem); this principle explains what our preprocessing approach accomplished. While it was expected that training a model in a federated learning fashion would be faster and yield higher accuracy on the transformed data, the additional benefit of being able to accomplish these results with a smaller model was uncovered during simulations led to our decision to break the norm of using models such as ResNet or VGG. Nonetheless, in this appendix section we introduce additional results and details regarding training a ResNet model in the same fashion detailed in this work (see A.2 for implementation details such as hyperparameters).

First and foremost, we now describe the details of the architecture of the ResNet. The model is composed of 8 convolutional layers and one, final, linear layer. Each convolutional layer contains a batch normalization step and a ReLu activation. There are two residual connections between a series of convolutional blocks and max pooling blocks which are used to scale the computed feature maps to the desired shape. Figure 6 shows the architecture step-by-step. The ResNet model contains approximately 6.5M parameters. It is nearly 6.2x larger than the custom convolutional neural network used in the body of work, which contains about 1M parameters.

During our simulations with ResNet, the benefit of achieving higher accuracy with fewer communication rounds, all while using a smaller model, is pronounced. Zooming in on the case where $N = P = 5$ (row one of table 3), using the smaller CNN took 35 rounds to reach 99.62% accuracy

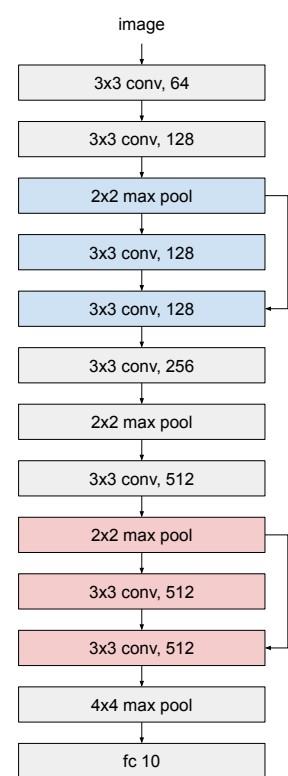

Figure 6: ResNet architecture.

Table 3: This table contains the results of simulations ran using the ResNet model.

| Client (N/P) | Communication Rounds | | Testing Accuracy | |
| --- | --- | --- | --- | --- |
| | ResNet | CNN | ResNet | CNN |
| 5/5 | 554 | 35 | 98.61 | 99.62 |
| 10/10 | 200 | 70 | 95.45 | 99.33 |
| 20/20 | 200 | 200 | 92.13 | 99.36 |

while it took the ResNet 554 communication rounds to max out at 98.61%. To compare, the ResNet, with 6.2x more parameters, required approximately 15.83x more communication rounds to reach 98.61% (1.01% less than the CNN). To ensure these results could be used for preliminary extrapolations of the minimum number of communication rounds until convergence of other simulations, we let the model be trained until its accuracy during training could not surpass a prior maximum within at least 100 communication rounds.

The results presented in table 3, are simulations using early stopping because the predicted minimum number of communication rounds needed to achieve equivalent testing accuracies obtained by the CNN were astoundingly larger. Using the multiple of 15.83x mentioned prior, we predict that for $N = P = 10$ and $N = P = 20$, it would take a minimum of $1,109$ and $3,166$ communication rounds, respectively, to achieve comparable results to the custom CNN. For these reasons, rows two and three of our table shows results that were stopped before full convergence. While some accuracy was still left on the table for those two cases, it still provides sufficient data to empirically demonstrate how using the smaller model was the better approach.

