# OpenReview forum: "Optimal Transport-Based Domain Alignment as a Preprocessing Step for Federated Learning"
_ICLR.cc/2025/Conference — Submitted to ICLR 2025_

### Official Review · Reviewer_gM2Z · 2024-10-28

**Soundness:** 1
**Presentation:** 1
**Contribution:** 2
**Rating:** 3
**Confidence:** 4

**Summary:**

This work proposes an optimal transport preprocessing step for federated learning in order to align the data distributions between the clients. The main idea is to treat the alignment as a hierarchical Wasserstein barycenters problem, where each client first summarizes their local data into an “average image” by computing a Wasserstein barycentre per image channel. These local barycenters are then communicated to the server, where new Wasserstein barycentres are computed across the local ones. These “global” barycenters are then communicated back to the clients and each one projects their data distribution onto them. After this preprocessing step, a standard federated learning algorithm such as FedAvg can be applied on top of these new input representations. The authors evaluate their method on CIFAR 10, showing improvements upon vanilla FedAvg and other baselines.

**Strengths:**

The main strength of this work is the simplicity of the proposed method. As it is a straightforward preprocessing step, it can be orthogonal to various other methods proposed for non-i.i.d. data in federated learning, potentially improving them even more.

**Weaknesses:**

I believe that this work has several axes for improvement.

# Clarity
There are quite a few instances where the manuscript is unclear and could be written better. More specifically:
- Important implementation details of the method are missing, i.e., what transport cost is used, what is the algorithm used for OT (e.g., is it Sinkhorn iterations?), what are its hyper parameters (if it is Sinkhorn iterations, what is the regularization parameter? This is important as it relates to the complexity analysis) and how is the projection to the global barycenters done. These need to be either in the main paper or in an appendix.
- The authors do not mention the dataset details in the experimental section (there is just a single mention of CIFAR-10 in the abstract) and what optimization hyper parameters where used for FedAvg. These are important for reproducibility and need to also be either in the main paper or the appendix.
- The notation in equations is a bit sloppy and needs to be improved. More specifically:
    - In Eq. 2 there is a $\sum_i$ but no index $i$ is present in the summand
    - Above Eq. 4 the definition of $(D^p_{i,j})_{i,j} \in R^{n\times n}$ is unclear; does this mean that each entry of the distance / cost matrix is itself an $n\times n$ matrix? Furthermore, there is a $diag(A) = 0$ introduced, but no mention of what $A$ is, and it is not clear what $\forall (i,j,k) [[n]]^3$ means and why it is important.
    - After Eq. 5 the authors optimise the Wasserstein barycentre $a \in \Sigma_n$ but there is no mention of what $\Sigma_n$ is.  Also it seems that also $\lambda_s \in \Sigma_n$ later on (after Eq. 7) which is a bit counterintuitive as one is a vector and the other is a scalar.
    - After Eq. 7 the authors mention $WB(B)$ but there is no mention of what $B$ is.

# Evaluation
In my opinion, there are quite a few ways that the evaluation can be improved:
- There seems to be only one dataset (assuming CIFAR-10 from the abstract?) and architecture used. This is not enough to get a strong signal for the usefulness of this method. Other datasets (such as TinyImagenet and CIFAR 100) and architectures (such as ResNets and ViTs) are important to provide a broader picture.
- It is not clear what the non-i.i.d.-ness is the data is. Uniform sampling without replacement should intuitively lead to more or less i.i.d. data across the clients, thus making OT-alignment not important. Prior works (such as McMahan et al. 2017) have more elaborate ways to generate non-i.i.d. data (by e.g., having each client only observe a subset of the classes). Furthermore, as the main premise of this work is domain alignment, I would also have expected a more elaborate evaluation of various non-i.i.d. settings, ranging from covariate shift (same $p(y)$ but different $p(x|y)$), to label skew (different $p(y)$ but same $p(x|y)$) or any mixture of the two. Intuitively, preprocessing the data should mainly help for the first setting as opposed to the second, so it would be beneficial if the authors update the manuscript with such cases.
- The final performance improvement is a bit hard to believe. The authors argue that their OT alignment step leads to ~ +28% better accuracy on their dataset against vanilla FedAvg (which is a very big improvement). If we take this claim at face value, then this would imply that this small architecture gets SoTA on CIFAR-10 (if I compare it with results shown in https://paperswithcode.com/sota/image-classification-on-cifar-10 where a much larger/deeper ViT-H/14 model gets 99.5%), which is also a bit hard to believe.

# Novelty
While the authors do discuss quite a few related works in the related work section, they unfortunately missed what seems to be a crucial related work, FedOT [1]. There the authors do also discuss about projecting the data distributions of each of the clients to a common space and they perform this step with optimal transport. Therefore, this weakens the novelty of this work.

[1] An Optimal Transport Approach to Personalized Federated Learning, Farnia et al., 2022, https://arxiv.org/abs/2206.02468

**Questions:**

The questions stem from the prior discussion of the weaknesses of this work. To reiterate:

# Clarity
- The authors should elaborate and extensively discuss the implementation details of the their method. Questions such as what transport cost are important and relevant for reproduction.
- The notation should be improved upon, with clear explanation of symbols.

# Evaluation
- The authors should increase the breadth and depth of the evaluation of their method, especially given its simplicity. Some recommendations are
    - OT-alignment combined with other federated learning methods
    - Evaluation on other datasets and architectures
    - More involved client data distribution shifts and how OT alignment improves performance
    - Perhaps a visual interpretation of what the Wasserstein barycenters look like
    - Clarification of how such a big performance improvement is possible with OT alignment. Maybe a relevant baseline is FedAvg on i.i.d. data splits, as that would highlight what is the best possible performance of OT alignment. Furthermore, are the numbers of the other baselines discussed at Table 2 taken from the respective papers or reimplemented? This is important, as hyperparameter details (especially with how the data are partitioned between the clients) can have a big impact on model behaviour.
    - The privacy claims are a bit vacuous, given that without formal privacy guarantees, FedAvg can break (e.g., see the work at [2]). I would encourage the authors to either substantiate the privacy claims (by e.g., an empirical evaluation) or removing them from the paper altogether.

[2] Inverting Gradients — How easy is it to break privacy in federated learning? Geiping et al., 2020, https://arxiv.org/abs/2003.14053

---

> ### Author Response · Authors · 2024-11-16
>
> First and foremost, thank you for your diligence in reviewing our work. We recognize the mistake we made regarding the statement on privacy. Here are the following steps we are taking to remedy the shortcomings:
>
> 1) We are adding a section in the Appendix which will cover all the implementation details, from hyperparameters to the details regarding the algorithm used to solve OT. We also make sure to include the details regarding what optimization algorithm was used to train FedAvg and the hyperparameter associated with it.
>
> 2) We have carefully passed over all mathematical notations to ensure clarity.
>
> 3) We agree that perhaps CIFAR-10 is too finely scoped. We chose this dataset because it is the most used dataset in the papers we came across. We did not use MNIST and F-MNIST because prior work did not have issues obtaining high accuracy scores on them and therefore would introduce no additional benefit. Regarding the architecture, we did not test more complex algorithms because we believed demonstrating the results we achieved with a smaller model was an additional benefit. We were not originally expecting these results, and after careful review of the training process to ensure no bias was added, we concluded that the results were of sufficient interest (smaller model yielded better results). In future work, we are looking to expand the complexity of the dataset and the model; perhaps even additional modalities of data as well.
>
> 4) We realized our approach of distributing the data did not follow conventional protocol (i.e., Dirichlet sampling). To this end we have remove the concept of non-i.i.dness and are making edits to the comparison values in the result table. While sampling without replacement can yield highly imbalanced datasets, as we saw in our simulations, we agree that because it did not follow standard methods that we should instead change our verbiage. We have made various updates regarding our language, result comparisons, and more.
>
> 5) As equally stated by the reviewer, we too initially second guessed the results. To ensure its accuracy we underwent several code reviews and in-depth analysis of the entire pipeline to double check that we did not introduce any bias. The results we compare our results against were using a smaller model. For this reason, the accuracy gap between the baseline FedAvg was larger than normal. For example, in the original work, the accuracy achieved on CIFAR-10 (with the larger model) was in the +80% range which lowers the gap from +20% to <20%. It is our belief that the alignment process is removing variance in the dataset which is making it easier to make predictions; the exact source of the reduced variance is unknown to us at this point (e.g., perhaps contrast reduction or color alignment). We believe that the projection step is simplifying finding the distinction between classes and reducing the complexity of the problem. For example, the kernel-trick to turn nonlinear, lower dimensional data to linearly separable in higher dimensions comes to mind when performing a transformation. Perhaps the variance reduced is reducing the nonlinearity of the underlying data and bringing similar samples together in the high dimensional space they live in. While we acknowledge that the accuracy score is SOTA, we do not agree a direct comparison with the listed methods (eg., ViT-H/14) is fair. Their algorithms are still learning over the original dataset and pay no additional cost (the preprocessing cost) in their pipelines. Running the architecture we used in our work over the original dataset does not yield the same accuracy as ViT-H/14 not does it yield close results to the ones presented in the paper.
>
> We again would like to thank the reviewer for the comments provided and would greatly appreciate feedback on our attempt to remedy our shortcomings. All comments are welcomed and if any additional concerns arise we would be thankful for the opportunity to remedy those as well.

---

### Official Review · Reviewer_GtqZ · 2024-11-04

**Soundness:** 3
**Presentation:** 2
**Contribution:** 3
**Rating:** 5
**Confidence:** 3

**Summary:**

The authors propose an optimal transport-based preprocessing step in federated learning to align datasets of different clients. This alignment step is run on each RGB color space separately using Wasserstein barycenters. After that, the datasets are projected toward the  target (aggregated) space. The authors show improvements over baselines empirically.

**Strengths:**

- The paper is well motivated and clearly written.

- As far as I know, use of optimal transport for domain alignment is novel in federated learning.

**Weaknesses:**

- The paper mentions several times that the proposed preprocessing algorithm would preserve privacy. However, I could not find a clear definition of privacy notion the paper is referring to. Could the authors clarify what privacy notion they're targeting and how they compute/measure the privacy guarantees?

**Questions:**

See above.

---

> ### Author Response · Authors · 2024-11-16
>
> First and foremost, thank you for your diligence in reviewing our work. We recognize the mistake we made regarding the statement on privacy. Our work is introducing a preprocessing framework that does not affect the learning process. Our intention was not to state we have improved privacy of FL and this was not clear in our writing. We have remedied this by removing any such statement and have reworded such statements to state our additions do not leak privacy. We have created an appendix to explain what we mean by “not breaking privacy constraints.” This statement is to merely assure the reader that when we share the barycenters, we are doing so in a privacy-retaining fashion. Because the computed barycenters are not invertible, meaning that we cannot obtain the data from the barycenters, we are not concerned with the privacy of the agent being leaked out. We again would like to thank the reviewer for the comments provided and would greatly appreciate feedback on our attempt to remedy our shortcomings.

---

### Official Review · Reviewer_xynD · 2024-11-04

**Soundness:** 2
**Presentation:** 1
**Contribution:** 2
**Rating:** 5
**Confidence:** 4

**Summary:**

This paper introduces a novel preprocessing step for federated learning that uses the tools of Optimal Transport. To summarize, it computes the Wasserstein barycenter of each client and collects them to compute a global barycenter. The clients then project local data to the global one in order to align the data with others. Experiments show significant improvement after the alignment compared with non-aligned baselines.

**Strengths:**

1. The author introduces a novel method using Optimal Transport to address the data heterogeneity in federated learning.
2. The method is impressive overall and seems easy to apply to different FL methods.

**Weaknesses:**

1. Overall presentation is below average: Many of the notations in section 3.2 are not explained. Also, the authors should focus on the meaning and utility of OT, instead of the formula derivation. Section 4 is lack of description. The authors should explain the whole process in detail by bullet list or paragraphs with subtitles based on Figure 2,3.
2. Weak Experiments: The comparisons are mostly with non-aligned FL methods but only one (CCVR) baseline. More baselines should be included to demonstrate OT to be a good preprocessing step.

**Questions:**

1. Please clarify the concepts and notations mentioned in Weakness 1 above. For example, section 3.2 introduced OT, but how WB was calculated is still unclear.
2. As noted in Weakness 2, the author mainly chooses non-aligned FL methods as the baseline. Could other preprocessing methods, like simple regularization, be included in the comparison? If not, why are further comparisons unavailable?

---

> ### Author Response · Authors · 2024-11-16
>
> First and foremost, thank you for your diligence in reviewing our work. We recognize the mistake we made regarding the statement on privacy. Here are the following steps we are taking to remedy the shortcomings:
>
> 1) We have fixed all the notations that had issues or were not properly explained. Section 4 has been expanded to better explain and describe the preprocessing work. The reason for a technical introduction of OT is due to comments in other publications asking for it.
>
> 2) We agree that further experiments could provide further baselines for comparison. In our findings, most papers overlap in using CIFAR-10 and this is the reason we chose to use it and establish an easy and direct comparison with other papers. We hope to be able to make much larger simulations in future research expanding to different models and data modalities as well.
>
> 3) During our research, we found only one preprocessing technique that was comparable. It was brought to our attention, and by complete accident we did not mention this work (which has already been remedied) was the work that introduced the FedOT algorithm which introduced a preprocessing idea. Other than their work, we did not find relevant points of comparison. Some ideas, as mentioned, include regularization and standardization. These techniques do not share information across agents that help images align. Further, some are already standard practice (e.g., MNIST is transformed from 0-255 to 0-1 by default).
>
> We again would like to thank the reviewer for the comments provided and would greatly appreciate feedback on our attempt to remedy our shortcomings.

---

### Official Review · Reviewer_DmHt · 2024-11-07

**Soundness:** 3
**Presentation:** 1
**Contribution:** 2
**Rating:** 5
**Confidence:** 3

**Summary:**

The paper proposes a preprocessing step for federated learning aimed to reduce distributional discrepancy between clients via optimal transport. It computes the channel-wise Wasserstein barycenters on each client and sends them to a trusted server where global Wasserstein barycenters are then computed as the target space and broadcasted to clients. Each client projects their data on the target space. Projected client data are fed into any FL learning algorithm. Experiments demonstrate that this preprocessing step pairing with FedAvg outperforms a number of baseline algorithms.

**Strengths:**

The proposed preprocessing step is interesting, simple yet effective in boosting FL learning performance.

**Weaknesses:**

Some concerns are as follows:
1) technical exposition in Section 3.2 is poor. For example, I have no idea what Lines 184-185 mean. There is no definition for $\mathcal{L}\_{d^P}(\cdot,\cdot)$ in Eq. (4), $\Sigma\_n$ in Eq.(5), $W\_{reg}$ in Eq. (7). Why $\lambda\_{s}\in\Sigma\_{n}$ in Line 215?
2) I feel confusing in Lines 339-350 regarding the number of epochs when the number of clients is small. In Lines 339-341, it seems to use a large number of epochs when P is small. But in Line 345 it seems to use a small number of epochs when P is small and in Line 349 it says that more agents less data.
3) In Line 469, could you explain why "we do so with a model of fewer parameters"?
4) it would be better if more experiments are presented where the preprocessing step pairs with more FL algorithms other than FedAvg.
5) The exposition of the proposed preprocessing step is instantiated with image data which has RGB channels. What if other data is given?
6) This preprocessing step relies on a trusted central server for privacy, which doesn't seem like a rigorous privacy guarantee.

**Questions:**

see above

---

> ### Author Response · Authors · 2024-11-16
>
> First and foremost, thank you for your diligence in reviewing our work. We recognize the mistakes pointed our work. Here is our plan to remedy it and we would appreciate feedback on these steps:
>
> 1) We have already corrected section 3.2 with its missing mathematical definitions and any additional mistakes.
>
> 2) For question number 2 regarding N and P, we have reworded what we wrote. Our goal was to explain the problem with making the number of total agents really high while keeping the number of participating agents small. For example, CIFAR-10 has 60K images. A network of 100 agents means each agent gets 600 images. If we only select 10 agents to participate in training, that means we only use 10% of the whole set of images. As the number increases we get smaller and smaller sets of images to use, making training harder, and in turn requiring more local training epochs.
>
> 3) The reason for using a smaller model was by happenstance. When coding our simulation, we used a controlled (small) model to be able to analyze every step of the code. When we achieved the results that we did, and double checked its validity to ensure there were no mistakes and biases, we decided to keep it. After all, we had better results with a smaller model and it seemed to us to present an additional benefit that we could demonstrate (better results, faster training time, higher accuracy with a smaller model).
>
> 4) We agree that we could have perhaps added the preprocessing step in various different algorithms. Our intention was to demonstrate the ability with the simplest algorithm used to solve FL. Various algorithms already exist that are better than FedAvg. If we can demonstrate the preprocessing improves FedAvg, the base for many algorithms, then follow up research can use it to improve in the areas where FedAvg lacks. For example, privacy is a big concern and differential privacy is one of the techniques used to increase privacy. Future research can use the technique we present and determine if it can allow differential privacy methods to increase noise while retaining utility.
>
> 5) While this work was limited to colored image, nothing prevents the same concept from being used in other areas. For example, grayscale images would repeat the process with only one barycenter. Furthermore, even though we have not done this yet for temporal data, the idea of transforming the data to some alternative space, or form, is not unfounded (e.g., Fourier transformation). We can see future work using the framework provided on various types of data.
>
> 6) Our work is merely introducing a preprocessing framework prior to any learning. Our intention was not to state we have improved privacy of FL and this was not clear in our writing. We have remedied this by removing any such statement. We have created an appendix to explain what we mean by “not breaking privacy constraints.” This statement is to merely assure the reader that when we pass the barycenters we do not leak privacy.
>
> We again would like to thank the reviewer for the comments provided and would greatly appreciate feedback on our attempt to remedy our shortcomings.

---

### Meta-Review · Area_Chair_edhj · 2024-12-21

**Metareview:**

In the paper, the authors proposed an Optimal Transport-based preprocessing algorithm that is designed to align datasets across edge devices by minimizing distributional discrepancies in the data. This approach ensures data consistency without compromising privacy.

After the rebuttal, there have remained several weaknesses of the paper: (1) The novelty of the proposed method is limited (e.g., the comparison with the related method, FedOT, is missing). (2) The reliability of the current experiments is not clear. Furthermore, some of the experiment results are not convincing. More experiments are needed to justify the proposed method. (3) Some of the major claims in the paper, such as the privacy claim, have remained vague.

Given the above weaknesses of the current paper, I recommend rejecting the paper at the current stage. The authors are encouraged to incorporate the suggestions and feedback of the reviewers into the revision of their manuscript.

**Additional Comments On Reviewer Discussion:**

Please refer to the meta-review.

---

### Decision · Program_Chairs · 2025-01-22

Reject